# The Model Sketch for Enhancing Lie Detection and Eliciting Information

**DOI:** 10.3390/brainsci12091180

**Published:** 2022-09-02

**Authors:** Haneen Deeb, Aldert Vrij, Sharon Leal, Samantha Mann, Jennifer Burkhardt

**Affiliations:** Department of Psychology, University of Portsmouth, Portsmouth PO1 2DY, UK

**Keywords:** deception, lie detection, model sketch, verbal cues, PLATO details

## Abstract

Background: Sketching while narrating is an effective interview technique for eliciting information and cues to deceit. The current research examined the effects of introducing a Model Sketch in investigative interviews andis pre-registered on https://osf.io/kz9mc (accessed on 18 January 2022). Methods: Participants (*N* = 163) completed a mock mission and were asked to tell the truth or to lie about it in an interview. In Phase 1 of the interview, participants provided either a free recall (control condition), sketched and narrated with exposure to a Model Sketch (Model Sketch-present condition), or sketched and narrated without exposure to a Model Sketch (Model Sketch-absent condition). In Phase 2, all participants provided a free recall without sketching. Results: Truth tellers reported significantly more information than lie tellers. The Model Sketch elicited more location details than a Free recall in Phase 1 and more veracity differences than the other Modality conditions in Phase 2. Conclusion: The Model Sketch seems to enhance the elicitation of information and to have carryover veracity effects in a follow-up free recall.

## 1. Introduction

Individuals differ physiologically when reporting a true versus false event with lie tellers showing greater activation in the prefrontal cortex than truth tellers [1]. However, observable (verbal and nonverbal) differences between lie tellers and truth tellers are negligible, which makes it difficult to detect deception by observation [2]. Previous research has shown that people detect lies at chance levels [3]. Therefore, interview techniques are currently under development aimed at increasing (mostly verbal) differences between lie tellers and truth tellers and enhancing lie detection [4]. One cluster of interview techniques encourage interviewees, particularly truth tellers, to talk more in order to elicit both new information and verbal cues to deceit [5,6]. One of the tools used in this cluster is sketching. It reinstates the context of an experience and therefore enhances memory and facilitates recall [7]. Another tool is the Model Statement, an example of a detailed verbal account unrelated to the event under investigation [8]. It raises interviewees’ understanding of how much information is required, thus increasing information elicitation [9]. These tools seem to benefit truth tellers more than lie tellers because lie tellers often prefer to report simple accounts [10,11]. In the current experiment, we combined the sketching and Model Statement tools by introducing a Model Sketch, an example of a detailed sketch. We examined its efficacy for eliciting information and cues to deceit in a sketch-based interview compared to an interview where such a Model Sketch is absent.

### 1.1. Sketching While Narrating in Investigative Interviews

Sketches are widely used during investigative interviews [12,13]. They are useful tools for understanding and visualising interviewees’ verbal accounts and for formulating interview questions [14]. Sketching has also been shown to be effective for eliciting accurate information and cues to deceit and for reducing memory contamination and suggestive questioning [15,16,17]. Sketching activates different brain regions (e.g., visual-sensory and prefrontal cortex) and enhances memory [18,19]. Sketching while narrating is more effective than a verbal free recall for eliciting information in truth tellers [20,21]. It mentally reinstates the context of the truth teller’s experience [22,23]. It is also more time consuming than a free recall, so by reinstating the context and allowing more time for retrieval, memory of activities relevant to the event are activated and enhanced [24,25,26]. Moreover, sketches are visual outputs and therefore more compatible with visually experienced events than a free recall, so they improve visual and spatial recall [27]. Further, sketching while narrating typically leads to the provision of spatial information as the interviewee must situate each person or object in a location on the sketch. In contrast, verbal recall does not require interviewees to spontaneously locate persons and objects [28].

Sketching while narrating therefore aids truth tellers’ memory which enables them to provide a richer account than if asked for a free recall. However, sketching while narrating is less likely to aid lie tellers’ accounts. Whereas both truth tellers and lie tellers want to enhance their self-presentation in the interview to appear honest, they accomplish this using different strategies [2]. Truth tellers demonstrate their honesty by reporting as many details as they can recall. In contrast, lie tellers are less willing to provide such information out of fear that the information gives leads to interviewers that they can check [29,30]. If lie tellers have not experienced the event, they are also less able to provide information because they may find it difficult to provide fabricated information that sounds plausible [31].

### 1.2. The Model Sketch in Investigative Interviews

In the current experiment, we examined how sketching while narrating can be made more effective for eliciting information and cues to deceit by introducing a Model Sketch in the interview. A Model Sketch is an example of a detailed sketch, adapted from the audiotaped Model Statement introduced by Leal et al. [8]. The Model Statement raises interviewees’ expectations of how much information they should provide and thus encourages them to offer more details [9,32]. A Model Statement is also an effective lie detection tool. Although it appears to generate a similar amount of additional information in truth tellers and lie tellers [33], it encourages truth tellers more than lie tellers to report specific types of information such as complications [34] and verifiable information [35]. Complications involve details that makes a story more complicated than necessary (e.g., I forgot to tie my shoes and thus I slipped off the ground). Truth tellers report more complications than lie tellers [36,37] because reporting complications goes against the lie tellers’ strategy of keeping their account simple [38]. Lie tellers also fear that reporting complications makes their account sound suspicious [39]. Further, truth tellers provide more verifiable sources (information that can be traced and checked such as receipts and phone calls) than lie tellers [40,41], because lie tellers do not want to give leads to the interviewer and verifiable sources can constitute evidence for or against an interviewee’s account.

When interviewees are asked for a sketch, it is more reasonable to expose them to an example of a sketch (Model Sketch) rather than to an audiotaped Model Statement so that the stimulus matches the visual output (the sketch they are asked to make). There is only one experiment to date that has examined the Model Sketch [42]. The experiment involved an interview with or without an interpreter present in which participants reported a true or false trip they have made. In the first phase of the interview, all participants were asked for a free recall. In the second interview phase, participants were first asked to note down keywords they plan to discuss after which they were either asked to provide a second free recall or to sketch and narrate with or without exposure to a Model Sketch. The Model Sketch comprised three detailed sketches of a dentist visit: one sketch of the building from the outside; one sketch of the waiting room; and one sketch of the dentist room. Significant differences emerged between truth tellers and lie tellers in both phases, but the Model Sketch did not elicit any significant effects. The authors explained that the absence of a Model Sketch effect may be the result of asking participants prior to introducing the sketch manipulation to note down keywords they plan to discuss. This may have committed participants to only report the keywords they noted down. The authors also alluded that the order of the recall and the sketch in the two interview phases (sketch always followed an initial free recall) may have affected the results.

The current experiment differed from Vrij et al. [42] in several important ways. First, participants were not asked to report any keywords prior to sketching. Second, participants in the sketching conditions were first asked to sketch and narrate (Phase 1) and then to provide a free recall without sketching (Phase 2). Third, dissimilar to Vrij et al.’s experiment, participants were given specific instructions encouraging them to provide as many details as were included in the Model Sketch. Fourth, Vrij and colleagues asked participants to report a trip they did or did not make. This means that lie tellers had to tell an outright lie. In daily life, however, lie tellers typically use embedded lies rather than outright lies in their reports [43,44]. Hence, we asked all of the participants to perform the same activity with lie tellers having to lie about some aspects of that activity (i.e., tell an embedded lie). Fifth, we presented participants with one sketch of the event (rather than three sketches) as we know from previous experience that participants typically use one paper for sketching a chain of events. Sixth, Vrij and colleagues carried out their interviews with or without the presence of an interpreter. We solely focused on the Model Sketch manipulation (without the presence of an interpreter) and compared three Modality conditions: Sketching while narrating with exposure to a Model Sketch (Model Sketch-present), sketching while narrating without exposure to a Model Sketch (Model Sketch-absent), and control (Free recall without sketching). Lastly, whereas Vrij and colleagues examined complications, common knowledge details, self-handicapping strategies, and total details, we focused on complications, verifiable sources, and PLATO (person, location, action, temporal, and object) details. Although PLATO details constitute the sum of details (i.e., total details) in an account [36,45], PLATO details allow for an assessment of the richness of the account which is more informative than assessing the quantity of information. This in turn allows the interviewer to formulate further questions based on those details which in turn enhances detectability [46,47,48] PLATO details have emerged as veracity cues in interviews involving sketches or a Model Statement [45,49].

### 1.3. Hypotheses

We predicted that truth tellers will be more likely than lie tellers to provide PLATO details, complications, and verifiable sources (Hypothesis 1, Veracity main effect).

We expected the same findings that emerged for the Model Statement to apply to the Model Sketch. Thus, participants in the Model Sketch-present condition will report the most PLATO details, complications, and verifiable sources and participants in the Free recall condition will report the least PLATO details, complications, and verifiable sources (Hypothesis 2, Modality main effect). We also predicted that the most profound differences between truth tellers and lie tellers will emerge in the Model Sketch-present condition and the least profound differences will emerge in the Free recall condition (Hypothesis 3, Veracity × Modality interaction effect).

We also examined whether the sketch was drawn from a first-person perspective or from an overhead perspective. Previous research has found that truth tellers provide sketches from a first-person perspective rather than from an overhead perspective [50]. Thus, we predicted that truth tellers will be more likely than lie tellers to provide sketches from a first-person perspective than from an overhead perspective (Hypothesis 4). 

These four hypotheses are registered on https://osf.io/kz9mc (accessed on 18 January 2022). In the pre-registration, we predicted effects for common knowledge details, but these did not emerge frequently enough, and the inter-rater reliability was extremely low (approaching zero), so we removed this variable from the analyses.

## 2. Materials and Methods

### 2.1. Participants and Design

A power analysis with G*Power software revealed that at least 162 participants are required to obtain a 95% statistical power, an alpha level of 0.05, and a medium to large effect size (*f*^2^ = 0.10), which is the effect found in previous deception research involving visuospatial tasks [28,51]. A total of 163 student and staff members (66% females) at the University of Portsmouth were recruited. Informed consent was obtained from all participants who took part. Age ranged between 18 and 65, and the average age was 24.30 (*SD* = 9.00). The sample included 63% Caucasians, 13% Asians, 10% Africans, 3% Hispanics, 3% Arabs, and 8% were of mixed ethnicity. Gender (all *F*’s ≤ 1.75, all *p*’s ≥ 0.177) and ethnicity (all *F*’s ≤ 2.16, all *p*’s ≥ 0.062) did not affect the results. Age had an effect in Phase 1 with older participants providing more PLATO details and complications than younger participants (all *t*’s ≥ 2.08, all *p*’s ≤ 0.039). Age had no effect on verifiable sources in Phase 1 (*t* = −0.18, *p* = 0.857) or on any of the dependent variables in Phase 2 (all *t*’s ≤ 1.82, all *p*’s ≥ 0.070). Hence, Age was included as a covariate in the Phase 1 analyses.

We carried out a between-subjects design with Veracity (truth, lie) and Modality (Model Sketch-present, Model Sketch-absent, Free recall) as factors and PLATO details, complications, verifiable sources, and sketch perspective as dependent variables. The sample included 85 truth tellers and 78 lie tellers. Among truth tellers, 28 participants were in the Model Sketch-present condition, 30 participants in the Model Sketch-absent condition, and 27 participants in the Free recall condition. Among lie tellers, 27 participants were in the Model Sketch-present condition, 24 participants in the Model Sketch-absent condition, and 27 participants in the Free recall condition.

### 2.2. Procedure

#### 2.2.1. Instructions

When they arrived at their appointment, participants read an instructions sheet about a mock mission they had to carry out to prevent an imminent cyberattack on a governmental agency. They were expected to collect and deliver a coded package with information about double agents involved in planning the attack. First, participants went to a room at the department to collect a bottle that would make them identifiable to the agents (confederates) from/to whom they collected/delivered the package. Participants then collected the package from an agent who was waiting at a shopping centre near the department. Next, they went to a tunnel in a nearby park to take a photo of the package and to send it to the mobile number of a second agent who was described as the Data Analyst to inform them that they were on their way to deliver the package. They then delivered the package to the Data Analyst in a room at the department. Similar missions were carried out in previous research [36,52].

Participants were given a tracking device to keep on them so that the experimenter can track the route they take and ensure that they take the route as instructed. However, participants were informed that the tracking device was for the experimenter to ensure that the mission was running smoothly.

After completing the mock mission and returning to the experimenter, participants were allocated to the Veracity and Modality conditions using block randomisation. Truth tellers were instructed as follows:

You will now be interviewed by an Agent about the mission. They will want to know everything about the mission, but they also need to be certain that you are telling the truth because there are other agents who may try to mislead them. Thus, please be truthful and tell the friendly agent everything you know about the mission.

Lie tellers were instructed as follows:

You will now be interviewed by an Agent about the mission. This person has intelligence clearance to know that you went on a mission today, and that an exchange happened in the Cascades, however, they do not know the exact details of what you did. We suspect that the Agent who will interview you is a double agent and as such we need you to lie about the true nature of your mission. It is important for us that you mislead them but at the same time appear cooperative so that they do not know you are suspicious about them. Therefore, you will have to lie and make up details about the following aspects of the mission: the content of the package, the Agent from whom you collected the package, and the Data Analyst to whom you delivered the package.

All participants were further instructed as follows:

You need to be convincing in the interview. If you are convincing, your name will be entered in a draw to win one of three prizes (£50, £100, £150). If you are not convincing, you will not be entered into the draw and you will be required to write an account of your mission, so please try your best. You may take as long as you need to prepare.

#### 2.2.2. The Interview

Two research assistants, blind to the study hypotheses and veracity conditions, interviewed participants. At the outset of the first interview phase, participants were asked to visualise details about the event:

I would like to know your activities for the past hour as I am collecting intelligence about an imminent cyberattack. Before I ask for your account, please take a few moments to picture in your mind the activities you have completed in the past hour. Think about where you were and what you saw, heard, felt and smelled each time. Take a moment to think about all your senses during those activities and then please let me know when you have done that.

Once ready, the interviewer asked participants in the Free recall condition to report in as much detail as possible everything they did and saw in the past hour. Participants in the Model Sketch-absent condition were provided with a blank, A3 white sheet of paper and asked as follows:

Please tell me in as much detail as possible everything you did and saw in the past hour. Whilst doing that, please sketch on this sheet of paper. Thus, you need to sketch and at the same time describe what you are sketching. You may use additional pieces of paper if you like, and you may take as long as you need to respond.

Participants in the Model Sketch-present condition were also presented with a blank, A3 white sheet of paper, in addition to two versions of a Model Sketch of the same event. The event was unrelated to the mission (fields, houses, and animals surrounding a main road with people around). The versions were from a first-person perspective and from an overhead perspective. We presented these versions so that the Model Sketch perspective does not affect the participant’s sketch perspective. The following instructions were provided:

Please tell me in as much detail as possible everything you did and saw in the past hour. Whilst doing that, please sketch on this sheet of paper. Thus, you need to sketch and at the same time describe what you are sketching. Before doing so, please look at this Model Sketch. It will give you an idea of how much detail I would like you to include in your response. So, try to include as many details as possible in your sketch and description. You may use additional pieces of paper if you like, and you may take as long as you need to respond.

Once participants responded, the interviewer left the room and returned back after five minutes to start the second interview phase in which all participants were asked for a verbal free recall about the mission.

#### 2.2.3. Post-Interview Questionnaire

After the interview, participants completed a post-interview questionnaire in which they rated on a percentage scale from 0 to 100 the extent to which they were truthful in the interview. They also rated on 7-point scales from 1 = *not at all* to 7 = *very much* the extent to which they (a) were motivated to appear convincing in the interview, (b) thought the interviewer believed them, (c) expected they would be entered in the prize draw, (d) expected they would have to write a statement, and (e) found the interview difficult. Participants were also asked an open question about the strategies they used to appear convincing during the interview. Participants who sketched also rated on 7-point scales from 1 = *not at all* to 7 = *very much* the extent to which sketching (a) helped them clarify what they needed to communicate, (b) encouraged them to provide details they would not have otherwise provided, and (c) enhanced the chances of them providing a convincing account. At the end of the experiment, participants read a debrief sheet and were thanked and remunerated with £20 or course credits for taking part.

#### 2.2.4. Coding

The interviews were transcribed and coded for each interview phase. The sketches were not coded, because a previous experiment [28] found no veracity differences in the sketches when participants sketched while narrating. Details that were repeated in a single phase were coded only once. In Phase 2, new PLATO details (i.e., details not mentioned during the first phase), new complications, and new verifiable sources were marked.

PLATO details were coded as person, location, action, temporal, or object details in line with previous PLATO coding schemes [49]. Person details involved the mention and physical descriptions of persons (e.g., “The agent was wearing a beige wristband” includes three person details). Location details referred to directions and to static places (e.g., streets and buildings) and their descriptions (e.g., “I passed the park opposite the train station” includes four location details). Action details were action verbs such as walked, entered, turned, passed, etc. Temporal details denoted time such as then, afterwards, before, throughout, etc. Object details referred to non-static objects such as monitors, cars, phones, and their descriptions (e.g., “she was holding a blue phone” includes two object details). The example “While she was walking, the streets smelled of weed. There were beggars around. She could see food in the shops” includes two person details (she, beggars), four location details (streets, around, in, shops), three action details (walking, smelled, see), one temporal detail (while), and two object details (weed, food).

We also coded complications and verifiable sources. The statement, “It was so cold and I wanted to wear my gloves but I could not because I needed my phone to do the mission” is an example of a complication. The statement, “I used my phone to send a photo of the package” includes two verifiable sources as the phone could be used to verify that a message was actually sent, and the photo could verify that the person had possession of the package.

The first author and a research assistant—both had previous experience in coding and were blind to the participants’ veracity and modality conditions—coded the transcripts independently. The first author coded all of the transcripts, and the research assistant coded 81 (50%) transcripts. Inter-rater reliability was measured with the Intra-Class Correlation (ICC) coefficient (single measures scores). The consensus is that inter-rater reliability is poor for ICC values less than 0.40, fair for values between 0.40 and 0.59, good for values between 0.60 and 0.74, and excellent for values between 0.75 and 1 [53]. Inter-rater reliability was good for temporal details (ICC = 0.73) and excellent for person details (ICC = 0.85), location details (ICC = 0.83), action details (ICC = 0.85), object details (ICC = 0.90), complications (ICC = 0.75), and verifiable sources (ICC = 0.86).

The two coders also coded the sketches for sketch perspective. The first-person perspective was defined as: ‘If you were standing at one angle (opposite the sketched scene), can you see all the scene/everything that is sketched?’. The ‘overhead’ perspective was defined as a sketch that can be seen from above, beyond what the participant is able to see. The first author rated all the sketches and the second coder rated 92 (84%) sketches. Inter-rater agreement was substantial, Cohen’s *κ* = 0.74.

Open responses on participants’ strategies to appear convincing were coded by the first author. A data driven approach was followed whereby categories were generated based on participants’ responses. Similar responses were grouped together in a single category. The selection of categories was grounded in previous research on truth tellers’ and lie tellers’ strategies [11]. When the same response could fit in more than one category, it was allocated to those categories. A second coder grouped the responses based on the corresponding categories generated by the first author. The first author explained the categories to the second coder and provided examples. Then, the second coder categorised a few responses that were checked and corrected by the first author. Afterwards, the second coder grouped all of the remaining responses to the corresponding categories. Inter-rater agreement was substantial, Cohen’s *κ* = 0.61.

## 3. Results

### 3.1. Post-Interview Questionnaire

We conducted a series of univariate analyses of variance (ANOVAs) with Veracity (lie teller, truth teller) and Modality (Model Sketch-present, Model Sketch-absent, Free recall) as factors and the variables mentioned in Table 1 as dependent variables (all measured on 7-point scales except for truthfulness which was measured on a percentage scale). Truth tellers reported significantly more truthful details than lie tellers, although lie tellers mentioned that approximately half of their statement was truthful (*M* = 46.92). Truth tellers were significantly more likely than lie tellers to think the interviewer believed them and that they would be entered in the prize draw, but less likely to believe that they would have to write a statement or that the interview was difficult. Truth tellers and lie tellers were equally motivated, and the average motivation was well above the midpoint of the scale.

Only one significant Modality effect emerged, for interview difficulty, *F*(2,57) = 3.25, *p* = 0.041, *η^2^* = 0.04. Post-hoc comparisons revealed that participants in the Model Sketch-present condition (*M* = 4.05, *SD* = 1.76, 95% CI [3.58, 4.53]) rated the interview as more difficult than those in the Free recall condition (*M* = 3.30, *SD* = 1.68, 95% CI [2.84, 3.75]). The Model Sketch-present and Free recall conditions did not differ from the Model Sketch-absent condition (*M* = 3.83, *SD* = 1.71, 95% CI [3.37, 4.30]). The other Modality main effects (all *F*s ≤ 2.24, all *p*s ≥ 0.110) and the Veracity × Modality interaction effects (all *F*s ≤ 2.11, all *p*s ≥ 0.125) were not significant.

We conducted another set of ANOVAs on data of participants who were asked to provide a sketch with Veracity and Modality as factors and the extent to which sketching (a) helped participants clarify what they needed to communicate, (b) encouraged them to provide details they would not have otherwise provided, and (c) enhanced the chances of them providing a convincing account (rated on 7-point scales) as dependent variables. None of the effects were significant, all *F*’s ≤ 3.93, all *p*’s ≥ 0.050.

The strategies used to appear convincing during the interview are displayed in Table 2. Truth tellers were more likely than lie tellers to provide a detailed account, to report the truth and to recall information from memory, to provide specific details such as describing specific people and locations, to offer a clear account, and to admit failures such as getting lost while doing the mission. Lie tellers, on the other hand, were more likely than truth tellers to keep their account simple by not providing many details, to report their account spontaneously and/or confidently, to control nonverbal and/or paraverbal behaviour, and to provide inaccurate information (deception). Approximately half of the lie tellers reported that they incorporated truthful details derived from the mission or from previous experience (i.e., told an embedded lie).

### 3.2. Hypotheses Testing

To check if we can conduct a multivariate analysis of variance to analyse our data, we ran a Pearson’s correlational analysis to test for multicollinearity between the dependent variables. Multicollinearity emerged for some of the variables (*r* > 0.80), so we decided to run univariate analyses of variance on the dependent variables. To control for the multiple testing problem, we divided the commonly used alpha level (*p* = 0.05) by the number of continuous dependent variables (0.05/7 = 0.007). Thus, only alpha levels below 0.007 were considered as significantly different from chance levels.

To test the interaction effects, we conducted simple orthogonal contrasts in each modality condition separately to compare truth tellers and lie tellers on each dependent variable.

The number of details provided in interview Phase 1 may have affected the number of new details provided in Phase 2. Therefore, we conducted univariate analyses of covariance (ANCOVAs) with the number of total details in Phase 1 as covariate when analysing Phase 2 data.

To corroborate our frequentist results, we carried out Bayesian analyses of (co)variance that test the likelihood of the data under both the null hypothesis (*H0*) and the alternative hypothesis (*H1*). Bayes factors (*BF*_10_) between 1 and 3 indicate weak evidence for the alternative hypothesis (*H1*) relative to the null hypotheses (*H0*), between 3 and 20 indicate positive evidence, between 20 and 150 indicate strong evidence, and above 150 indicate very strong evidence [54]. A Bayes factor close to 1 means no evidence can be derived from the data for either the null or the alternative hypothesis. The inverse of *BF*_10_ is *BF*_01_ (1/*BF*_10_), which is the likelihood of supporting evidence for the null hypothesis (*H0*) compared to the alternative hypothesis (*H1*). We report *BF_10_* statistics only in Table 3, Table 4 and Table 5 as *BF*_01_ can be inferred by inversing *BF*_10_.

### 3.3. Verbal Cues in Interview Phase 1

To test Hypotheses 1 and 2 in Phase 1, we conducted a series of 2 (Veracity: lie teller, truth teller) × 3 (Modality: Model Sketch-present, Model Sketch- absent, Free recall) ANCOVAs with PLATO details, complications, and verifiable sources in Phase 1 as dependent variables and Age as covariate. The Veracity main effects are shown in Table 3. Truth tellers provided significantly more PLATO details, complications, and verifiable sources than lie tellers. The effect sizes ranged between 0.48 and 1.58, and the Bayesian analyses showed positive to very strong evidence for the alternative hypothesis relative to the null hypothesis. Thus, Hypothesis 1, that predicted Veracity main effects for all dependent variables, was supported in Phase 1.

A significant Modality effect emerged for location details, *F*(2,156) = 7.14, *p* = 0.001, *η*^2^ = 0.08, BF_10_ = 7.18. More location details emerged in the Model Sketch-present condition (*M* = 63.22, *SD* = 42.34, 95% CI [51.77, 74.66]) than in the Free recall condition (*M* = 40.96, *SD* = 31.82, 95% CI [32.28, 49.65]). The Model Sketch-present and Free recall conditions did not differ from the Model Sketch-absent condition (*M* = 48.91, *SD* = 30.07, 95% CI [40.70, 57.11]). Thus, Hypothesis 2, that predicted a Modality main effect, was supported for location details only in Phase 1.

To test the interaction effect in Phase 1 (Hypothesis 3), we conducted simple orthogonal contrasts in each Modality condition by comparing truth tellers and lie tellers on each dependent variable. The results can be found in Table 4. In the Model Sketch-present condition, truth tellers provided significantly more action details, temporal details, and verifiable sources than lie tellers, and these results were supported by the Bayesian analyses. The Bayesian analyses also revealed positive evidence that truth tellers provided more complications than lie tellers. The effect sizes for these four variables ranged between medium and large (0.71 ≤ *d* ≤ 2.15), and the average effect size was 1.09.

In the Model Sketch-absent condition, truth tellers provided significantly more action details, temporal details, and verifiable sources than lie tellers. The Bayesian analyses supported these results and also provided evidence in favour of veracity differences for object details. The effect sizes for these variables were medium to large (0.76 ≤ *d* ≤ 1.32) and averaged 1.07.

In the Free recall condition, truth tellers provided significantly more PLATO details, complications, and verifiable sources than lie tellers, and these results were supported by the Bayesian analyses. The effect sizes were large (0.94 ≤ *d* ≤ 2.04) and averaged 1.32. Overall, the findings suggest that the Veracity differences were large in each of the three interview conditions and perhaps particularly pronounced in the Free recall condition. This does not provide support for Hypothesis 3 in Phase 1.

### 3.4. Verbal Cues in Interview Phase 2

To test Hypotheses 1 and 2 in Phase 2, we ran a series of 2 (Veracity: lie teller, truth teller) × 3 (Modality: Model Sketch-present, Model Sketch- absent, Free recall) ANCOVAs on new PLATO details, new complications, and new verifiable sources in Phase 2 with total number of details in Phase 1 as covariate. The Veracity main effects (all *F*’s ≤ 5.82, all *p*’s ≥ 0.017) and the Modality main effects (all *F*’s ≤ 1.58, all *p*’s ≥ 0.209) were not significant. However, the Bayesian analysis for the Veracity effect showed positive evidence for new action details (see Table 3). Thus, Hypothesis 1 was partially supported, and Hypothesis 2 was not supported in Phase 2.

The simple orthogonal contrasts that tested Hypothesis 3 in Phase 2 are shown in Table 5. In the Model Sketch-present condition, truth tellers provided significantly more new action details than lie tellers, and the Bayesian analysis showed positive evidence in favour of this result. The effect size was large (*d* = 1.03). No significant effects emerged for the Model Sketch-absent condition or the Free recall condition, and the Bayesian analyses did not show evidence in favour of the alternative hypothesis. The findings thus suggest that the most pronounced Veracity differences were found in the Model Sketch-present condition, partially supporting Hypothesis 3 in Phase 2.

### 3.5. Sketch Perspective

We ran a three-way loglinear analysis to test sketch perspective (Hypothesis 4) with Veracity, Modality, and Sketch perspective as factors. No significant k-way effects emerged (all $\chi^{2}$ ≤ 11.56, all *p’*s ≥ 0.082). However, the parameter estimates showed a significant effect of Sketch perspective revealing that more sketches were provided from a first-person perspective (62%) than from an overhead perspective (38%), Z = 2.43, *p* = 0.015. Thus, Hypothesis 4 that predicted a Veracity effect was not supported.

## 4. Discussion

### 4.1. Veracity Main Effects

Truth tellers provided significantly more PLATO details, complications, and verifiable sources than lie tellers in Phase 1. This finding is consistent with previous lie detection research demonstrating that truth tellers provide richer accounts and are more willing and able to provide information than lie tellers [36,45,55,56]. The results are also consistent with neurological approaches to deceptive behaviours showing that truth tellers and lie tellers differ in how they process their reports [1]. The convincing strategies reported by truth tellers and lie tellers (see Table 2) also corroborate these findings as truth tellers mentioned providing detailed and specific information and admitting failures (which are usually coded as complications) more than lie tellers. In contrast, lie tellers were more likely to use a ‘keep it simple’ strategy by not providing many details. This shows that although lie tellers try to present themselves positively before the interviewer, they do this in a different way than truth tellers, which in turn enhances veracity differences [2,57].

Approximately half of the lie tellers mentioned that they included embedded lies. This may have resulted from our instructions to lie tellers that they can use truthful information as long as they mislead the interviewer concerning specific parts of the mission. We produced these instructions based on previous findings that lie tellers are more likely to tell embedded lies than outright lies and thus incorporate truthful details in their account [43,44]. Even when they impart truthful information, the neurological processes and the quality of lie tellers’ accounts are different from those of truth tellers [1,58,59,60]. Our results confirmed that these veracity differences continue to emerge even when lie tellers include a large proportion of truthful details in their account.

In Phase 2, one Veracity effect emerged: truth tellers provided significantly more new action details than lie tellers. This again resonates with previous research that truth tellers are more likely than lie tellers to provide new information in a follow-up interview [61,62]. However, our findings were limited to new action details. It may be that truth tellers could not provide more new information related to other PLATO details, complications, and verifiable sources due to a ceiling effect. That is, truth tellers may have provided all of the information they could provide in Phase 1, leaving little new information to be added in Phase 2; this ultimately reduced the differences between truth tellers and lie tellers in Phase 2.

### 4.2. Modality Main Effects

Only one Modality effect emerged: the Model Sketch yielded more location details than a Free Recall in Phase 1. Thus, the Model Sketch seems to work similar to a Model Statement by giving interviewees an example of and guiding them on how much information they should provide. The finding that the Model Sketch enhanced the elicitation of location details in the interview phase in which it was introduced has practical implications. In real life interviews, this information is often solicited and deemed important for collecting more intelligence relevant to the target site and for verifying the interviewee’s account (Mark Fallon, ClubFed LLC, Brunswick, Georgia, USA. Personal communication, 2018). In such cases, presenting a Model Sketch seems to be a valid technique for enhancing the provision of location details.

The Modality effect for location details did not carry over to Phase 2. Hence, it seems that the advantage of the Model Sketch as an information elicitation tool is limited to the interview in which it is introduced. This finding aligns with previous research on the Model Statement and sketching while narrating where these tools enhanced information elicitation in the interview in which they were introduced but did not show carryover effects [49,61,63].

The Model Sketch enhanced the provision of information related to the spatial setting only. The scenario used in the current experiment may have affected the results as participants could report a lot of location information relevant to the route taken. This raises the question of whether the Model Sketch is context dependent. For example, had the scenario included more persons and complications, would participants exposed to the Model Sketch report more person details and complications than their counterparts in a Free recall? Future research could address this question.

### 4.3. Veracity × Modality Interaction Effect

The Model Sketch did not result in larger Veracity effects than the Free Recall condition in Phase 1. This finding contradicts previous research which has shown that sketching enhances lie detection more than a free recall [17,28]. The Veracity effects were very large in all Modality conditions in Phase 1 (*d* > 1), including the Free Recall condition. Hence, there seems to be a ceiling effect whereby the Model Sketch could not further enhance the veracity effects. Similar findings emerged in previous research where large effects in a Free recall condition limited the effects of interview techniques aimed at enhancing differences between truth tellers and lie tellers [49].

The Model Sketch was the only Modality condition that showed a significant and large Veracity effect in Phase 2 (for new details) after controlling for total details in Phase 1. Since the Model Sketch was only introduced in Phase 1, this Veracity effect can be interpreted as a carryover effect. It has been found in previous research that Veracity effects of a Model Statement alone or sketching while narrating alone do not carry over to subsequent interviews but are limited to the interviews in which these tools were introduced [61,63]. The current research shows that combining the two techniques by introducing a Model Sketch may instigate such carryover Veracity effects.

Distinct from the null findings by Vrij et al. [42], we found that the Model Sketch had a large effect in the interview in which it was introduced, and its effects carried over to a follow-up interview. We cannot explain the discrepant results in both experiments because there were many differences between them. Our results, however, suggest that the Model Sketch is a more promising information elicitation and lie detection tool than initially found and that it works similar to a Model Statement by encouraging interviewees to provide information. The Model Statement has been well received by practitioners and is employed in practical settings given its effectiveness and efficiency [9]. The present experiment shows that the Model Sketch may be used as an alternative in sketch-based interviews, but more research is needed to understand the contexts in which it works and which verbal cues it elicits.

### 4.4. Sketch Perspective

The majority of participants sketched from a first-person perspective rather than from an overhead perspective. This does not support previous research in which truth tellers, more than lie tellers, sketched from a first-person perspective [50]. The instructions given to participants may explain the discrepancies in findings. In Vrij et al. [50], lie tellers were asked to lie about the location and sketched a different location than where they experienced the event (an exchange of a package with another agent). This did not happen in the present experiment where lie tellers were truthful about the location where the exchange took place and could thus sketch from a first-person perspective similar to truth tellers.

### 4.5. Limitations and Future Directions

We formulated hypotheses concerning common knowledge details in the pre-registration, but these cues did not occur frequently enough in the current experiment. Given that participants (both truth tellers and lie tellers) could incorporate truthful details from personal experience, they were able to report unique experiences and did not need to rely on scripted information that is commonly known to everyone.

We predicted that we would find medium to large effects, and based on these predictions, we recruited participants with cell sizes ranging between 24 and 30. Nonetheless, we found small effect sizes which may nonetheless be important in real life interviews. Hence, our sample size may not be large enough to detect all small effect sizes of a Model Sketch or to allow us to generalise the findings. A larger sample size may have provided more insight into these small effects.

We used one scene (a countryside with farms, houses, animals, and people) for the Model Sketch in the current experiment. In theory, a Model Sketch with a different content could have affected the results. However, we do not expect to find any differences as long as the content is unrelated to the event under investigation (the mission in our experiment).

We did not code sketches as previous research showed that truth tellers’ and lie tellers’ sketches did not differ when they sketched and narrated [28]. However, more research is needed to examine whether the sketching component alone could elicit veracity differences in a sketching while narrating interview in the presence of a Model Sketch.

We examined specific verbal cues to deception that have been validated in research involving sketching while narrating interviews. Researchers have tested different verbal cues (e.g., plausibility, consistency) in interviews involving a sketch only (without narration) and these were shown to reliably distinguish truths from lies [50,64,65]. Future research may investigate if these verbal cues can also be elicited when interviewees sketch and narrate in the presence of a Model Sketch.

In the current research, coders rated the interviews for verbal cues. An alternative lie detection method is to ask judges to read the transcripts and to decide whether they are true or false based on those specific verbal cues. A meta-analysis of previous research using judges has shown that judges who were instructed to look for valid verbal cues achieved a lie detection accuracy rate of 76% [66].

## 5. Conclusions

We attempted to replicate the first experiment that tested the Model Sketch [36] which did not find supporting evidence. Differing from the original experiment, we employed manipulations that focused solely on the Model Sketch. We found that the Model Sketch may be a promising lie detection and information elicitation tool in interviews in which interviewees are asked to sketch and narrate. As with the Model Statement, it appears to guide interviewees on how much information they should provide. The Model Sketch was particularly effective for eliciting information on locations. It also elicited veracity differences, but it did not outperform a free recall. However, in a follow-up free recall interview, the Model Sketch showed a carryover effect that distinguished truth tellers and lie tellers beyond the free recall and sketching while narrating interviews in which participants were not exposed to the Model Sketch.

## Figures and Tables

**Table 1 brainsci-12-01180-t001:** Descriptive and Inferential Statistics for the Ratings in the Post-Interview Questionnaire.

Questionnaire Item	Truth Tellers		Lie Tellers		F	*p*	*η* ^2^
	M (SD)	95% CI	M (SD)	95% CI			
Truthfulness	98.59 (05.38)	97.40, 99.75	46.92 (23.92)	41.52, 52.61	371.83	<0.001	0.70
Motivation	5.72 (1.57)	5.37, 6.06	5.33 (1.20)	5.06, 5.61	3.09	0.081	0.02
Believed by interviewer	5.11 (1.49)	4.78, 5.43	4.01 (1.55)	3.71, 4.42	21.07	<0.001	0.12
Prize draw	4.48 (1.85)	4.08, 4.89	3.77 (1.83)	3.39, 4.24	6.46	0.012	0.04
Writing a statement	3.13 (1.63)	2.78, 3.48	4.07 (1.67)	3.68, 4.45	12.64	<0.001	0.08
Interview difficulty	3.27 (1.64)	2.93, 6.34	4.23 (1.71)	3.79, 4.58	14.15	<0.001	0.08

**Table 2 brainsci-12-01180-t002:** Frequency of Convincing Strategies as Reported by Truth Tellers and Lie Tellers.

Strategies	Truth Tellers	Lie Tellers
Provided a detailed account	48	2
Reported the truth and/or recalled information	42	0
Provided specific details	24	13
Controlled nonverbal and/or paraverbal behaviour	11	15
Offered a clear account	8	2
Reported spontaneously and/or confidently	7	16
Maintained consistency	7	9
Visualised/Imagined the event	6	5
Mentioned feelings and/or thoughts	5	4
Admitted failures	5	0
Kept it simple/Did not provide many details	2	28
Incorporated truthful details within a false account	0	34
Provided inaccurate information	0	12

**Table 3 brainsci-12-01180-t003:** Descriptive and Inferential Statistics for the Dependent Variables as a Function of Veracity in Phases 1 and 2.

Detail Type	Truth Tellers *n* = 85		Lie Tellers *n* = 78		F	*p*	*d*	BF_10_
	M (SD)	95% CI	M (SD)	95% CI				
**Phase 1**								
Person details	11.07 (10.36)	8.84, 13.31	6.63 (7.87)	4.85, 8.40	9.12	0.003	0.48 [0.17, 0.80]	11.92
Location details	62.98 (36.72)	55.06, 70.90	38.17 (30.99)	31.18, 45.15	23.20	<0.001	0.73 [0.41, 1.05]	2289.20
Action details	24.26 (10.31)	22.04, 26.48	14.31 (7.62)	12.59, 16.03	53.57	<0.001	1.10 [0.77, 1.43]	9.560 × 10^7^
Temporal details	21.48 (13.11)	18.65, 24.31	10.56 (8.26)	8.70, 12.43	39.75	<0.001	1.00 [0.67, 1.32]	3.396 × 10^6^
Object details	13.26 (9.67)	11.17, 15.35	6.44 (6.18)	5.04, 7.83	29.05	<0.001	0.84 [0.52, 1.16]	36743.64
Complications	1.15 (1.68)	0.79, 1.52	0.31 (0.74)	0.14, 0.48	16.60	<0.001	0.65 [0.33, 0.96]	297.27
Verifiable sources	2.94 (1.48)	2.62, 3.26	0.85 (1.15)	0.59, 1.11	101.11	<0.001	1.58 [1.22, 1.93]	3969 × 10^15^
**Phase 2**								
New person details	4.93 (11.61)	2.42, 7.43	2.82 (4.38)	1.83, 3.81	0.06	0.806	0.24 [−0.07, 0.55]	0.18
New location details	24.85 (32.82)	17.77, 31.93	13.21 (13.20)	10.23, 16.18	0.03	0.867	0.47 [0.15, 0.78]	0.19
New action details	6.51 (5.97)	5.22, 7.79	3.35 (3.26)	2.61, 4.08	5.82	0.017	0.66 [0.34, 0.97]	3.62
New temporal details	6.59 (7.39)	4.99, 8.18	2.56 (2.95)	1.90, 3.23	3.61	0.059	0.72 [0.40, 1.04]	1.16
New object details	4.72 (7.73)	3.05, 3.69	2.24 (2.93)	1.58, 2.90	0.38	0.536	0.42 [0.11, 0.74]	0.23
New complications	0.32 (0.69)	0.17, 0.47	0.15 (0.67)	0.00, 0.30	0.44	0.508	0.25 [−0.06, 0.56]	0.26
New verifiable sources	0.54 (0.93)	0.34, 0.74	0.29 (0.76)	0.12, 0.47	2.24	0.136	0.29 [−0.02, 0.61]	0.60

**Table 4 brainsci-12-01180-t004:** Descriptive and Inferential Statistics for the Dependent Variables as a Function of Veracity and Modality in Phase 1.

Detail Type	Truth Tellers		Lie Tellers		F	*p*	*d*	BF_10_
	M (SD)	95% CI	M (SD)	95% CI				
**Model Sketch-present**								
Person details	10.00 (7.09)	7.25, 12.75	8.56 (10.65)	4.34, 12.77	0.52	0.473	0.16 [−0.38, 0.70]	0.32
Location details	73.25 (44.76)	55.89, 90.61	52.81 (37.70)	37.90, 67.73	4.12	0.047	0.49 [−0.05, 1.04]	1.07
Action details	23.29 (9.71)	19.52, 27.05	16.78 (7.90)	13.65, 19.90	8.32	0.006	0.74 [0.18, 1.29]	5.28
Temporal details	22.32 (12.14)	17.62, 27.03	13.78 (10.02)	9.81, 17.74	8.12	0.006	0.77 [0.21, 1.33]	6.83
Object details	14.50 (12.34)	9.72, 19.28	7.70 (8.12)	4.49, 10.92	6.67	0.013	0.65 [0.10, 1.20]	2.81
Complications	1.00 (1.47)	0.43, 1.57	0.22 (0.51)	0.02, 0.42	6.90	0.011	0.71 [0.15, 1.26]	4.21
Verifiable sources	2.75 (0.97)	2.38, 3.12	0.74 (0.90)	0.38, 1.10	61.37	<0.001	2.15 [1.47, 2.83]	5.203 × 10^7^
**Model Sketch-absent**								
Person details	11.17 (13.69)	6.05, 16.28	6.21 (6.33)	3.53, 8.88	2.55	0.116	0.47 [−0.09, 1.02]	0.83
Location details	57.17 (30.39)	45.82, 68.52	38.58 (26.80)	27.27, 49.90	5.63	0.021	0.65 [0.09, 1.21]	2.56
Action details	21.40 (8.26)	18.32, 24.48	12.33 (5.06)	10.20, 14.47	22.83	<0.001	1.32 [0.72, 1.93]	951.91
Temporal details	20.43 (12.54)	15.75, 25.12	9.50 (6.10)	6.92, 12.08	15.42	<0.001	1.11 [0.52, 1.70]	94.40
Object details	11.47 (8.44)	8.32, 14.62	6.12 (5.36)	3.86, 8.39	7.66	0.008	0.76 [0.19, 1.32]	5.00
Complications	1.17 (2.17)	0.36, 1.98	0.50 (0.83)	0.15, 0.85	2.32	0.134	0.41 [−0.14, 0.96]	0.63
Verifiable sources	2.77 (1.92)	2.05, 3.49	0.96 (1.43)	0.35, 1.56	15.50	<0.001	1.07 [0.48, 1.65]	76.47
**Free recall**								
Person details	12.07 (09.12)	8.47, 15.68	5.07 (5.28)	2.98, 7.16	8.48	0.005	0.94 [0.37, 1.51]	28.28
Location details	58.78 (32.77)	45.82, 71.74	23.15 (18.19)	15.95, 30.34	20.47	<0.001	1.34 [0.74, 1.95]	1928.84
Action details	28.44 (11.85)	23.76, 33.13	13.59 (8.74)	10.13, 17.05	22.88	<0.001	1.43 [0.82, 2.04]	4955.95
Temporal details	21.78 (15.00)	15.84, 27.71	8.30 (7.19)	5.45, 11.14	14.58	<0.001	1.15 [0.56, 1.73]	218.95
Object details	13.96 (07.68)	10.93, 17.00	5.44 (4.34)	3.73, 7.16	20.78	<0.001	1.37 [0.76, 1.97]	2473.64
Complications	1.30 (01.27)	0.80, 1.80	0.22 (0.85)	−0.11, 0.56	11.70	0.001	1.00 [0.42, 1.58]	49.01
Verifiable sources	3.33 (01.30)	2.82, 3.85	0.85 (1.13)	0.40, 1.30	48.89	<0.001	2.04 [1.36, 2.71]	8.709 × 10^6^

**Table 5 brainsci-12-01180-t005:** Descriptive and Inferential Statistics for the Dependent Variables as a Function of Veracity and Modality in Phase 2.

Detail Type	Truth Tellers		Lie Tellers		F	*p*	*d*	BF_10_
	M (SD)	95% CI	M (SD)	95% CI				
**Model Sketch-present**								
New person details	4.07 (4.50)	2.33, 5.81	3.33 (4.75)	1.46, 5.21	0.21	0.649	0.16 [−0.38, 0.70]	0.29
New location details	28.11 (24.42)	18.64, 37.58	13.19 (12.00)	8.44, 17.93	2.84	0.098	0.78 [0.22, 1.33]	0.98
New action details	6.71 (4.35)	5.03, 8.40	3.07 (2.50)	2.09, 4.06	9.52	0.003	1.03 [0.45, 1.60]	15.77
New temporal details	7.11 (6.76)	4.48, 9.73	2.89 (3.33)	1.57, 4.21	3.50	0.067	0.79 [0.23, 1.35]	1.30
New object details	5.25 (5.07)	3.28, 7.22	2.26 (2.65)	1.21, 3.31	2.88	0.096	0.74 [0.18, 1.30]	1.04
New complications	0.29 (0.71)	0.01, 0.56	0.04 (0.19)	−0.04, 0.11	2.12	0.151	0.48 [−0.07, 1.03]	0.71
New verifiable sources	0.64 (1.06)	0.23, 1.05	0.15 (0.46)	−0.03, 0.33	3.40	0.071	0.60 [0.05, 1.15]	1.26
**Model Sketch-absent**								
New person details	4.03 (4.80)	2.24, 5.83	2.08 (4.79)	0.06, 4.11	1.05	0.311	0.41 [−0.15, 0.96]	0.47
New location details	18.63 (12.71)	13.89, 23.38	11.21 (13.78)	5.39, 17.03	1.15	0.289	0.56 [0.00, 1.12]	0.53
New action details	5.63 (3.79)	4.22, 7.05	3.38 (3.44)	1.92, 4.83	4.08	0.049	0.62 [0.06, 1.18]	1.66
New temporal details	5.27 (4.06)	3.75, 6.78	2.00 (2.83)	0.81, 3.19	4.83	0.033	0.93 [0.36, 1.51]	2.67
New object details	2.63 (2.14)	1.83, 3.43	2.46 (3.75)	0.87, 4.04	0.14	0.712	0.06 [−0.49, 0.60]	0.30
New complications	0.33 (0.66)	0.09, 0.58	0.12 (0.45)	−0.06, 0.31	0.95	0.334	0.37 [−0.18, 0.92]	0.48
New verifiable sources	0.53 (0.97)	0.17, 0.90	0.21 (0.51)	−0.01, 0.42	3.97	0.052	0.41 [−0.14, 0.97]	1.26
**Free recall**								
New person details	6.81 (19.58)	−0.93, 14.56	2.96 (3.62)	1.53, 4.40	1.02	0.318	0.27 [−0.27, 0.82]	0.43
New location details	28.37 (51.10)	8.16, 48.58	15.00 (14.03)	9.45, 20.55	1.18	0.282	0.36 [−0.19, 0.90]	0.42
New action details	7.26 (8.83)	3.77, 10.75	3.59 (3.83)	2.08, 5.11	0.03	0.865	0.54 [−0.01, 1.09]	0.32
New temporal details	7.52 (10.37)	3.42, 11.62	2.74 (2.68)	1.68, 3.80	0.00	0.963	0.63 [0.07, 1.19]	0.34
New object details	6.48 (12.36)	1.59, 11.37	2.04 (2.41)	1.08, 2.99	0.03	0.859	0.50 [−0.05, 1.05]	0.32
New complications	0.33 (0.73)	0.04, 0.62	0.30 (1.03)	−0.11, 0.70	0.68	0.413	0.03 [−0.51, 0.58]	0.36
New verifiable sources	0.44 (0.75)	0.15, 0.74	0.52 (1.09)	0.09, 0.95	0.54	0.465	0.09 [−0.46, 0.63]	0.36

## Data Availability

The data can be accessed on https://osf.io/w9c4d/ (accessed on 18 January 2022).

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
