# Peer review of "The Model Sketch for Enhancing Lie Detection and Eliciting Information"

_brainsci, 2022, doi:10.3390/brainsci12091180_

Round 1
Reviewer 1 Report
Review of Manuscript #brainsci-1884466 submitted to Brain Sciences:
The model sketch for enhancing lie detection and eliciting information
August 20th, 2022
The purpose of this submission was to study the effect of requiring participants to complete a diagram, or “Model Sketch,” while either telling the truth or lying about a mock anti-terrorist mission they completed. In a between-participants design, participants either told the truth or lied about their mission, and they sketched and narrated their activity while looking at a sample Model Sketch, sketched and narrated without exposure to the sample Model Sketch, or narrated their activity without completing a sketch. In a second phase of the study, all participants narrated their activity without sketching. Across the sketching conditions, truthful participants reported more information than deceptive participants. Sketching and narrating while looking at a Model Sketch caused participants to report more details about their mock mission.
This research extends knowledge regarding the well known “investigative interview” techniques by Vrij and colleagues for detecting deception. While only some of the hypotheses were supported, the results indicate that the use of Model Sketches may be a promising avenue for distinguishing between truthful and deceptive communications under some circumstances. The fact that the study was pre-registered adds confidence to the findings.
Most importantly, I believe this manuscript could benefit from clarifying some points about the predictions, as well as inserting a few important additions to the background literature.
As mentioned earlier, the authors should be lauded for pre-registering their study, which reduces the likelihood of false positives and prevents researchers from intentionally or unintentionally presenting post-hoc hypotheses as a priori hypotheses (among other benefits). With that in mind, there are a few points in this manuscript where the authors could clarify which statistical tests were based on the pre-registered hypotheses and which tests were exploratory. In particular, it is not clear if all of the findings on p. 7 of the manuscript were based on exploratory analyses or if they were part of the four main hypotheses. Also, were all of the Bayesian analyses pre-registered? Were the ANCOVAs mentioned on p. 10 (section 3.4) pre-registered? In the Results and Discussion section, the authors could differentiate more explicitly between hypothesis testing vs. exploratory analyses.
The authors are experts in the field of deception detection, and they provide ample background on their excellent research in this field. However, I believe that the introduction and discussion could benefit from some discussion of research by other deception detection experts, using different paradigms. For example, when the authors discuss characteristics of lie tellers generally on p. 2, it would be helpful to cite meta-analyses by Bella DePaulo and/or research by Tim Levine on this topic. The same suggestion applies to the early Discussion section (p. 11), when the authors discuss differences between liars and truth tellers in previous research.
I have a question regarding the effects related to Hypothesis 3, as discussed on p. 10 of the manuscript. The authors noted that truth tellers provided significantly more new action details than liars in the Model Sketch-present condition, and then point out that there were no significant effects in the Model Sketch-absent condition. The implication here is that the presence of a Model Sketch sample leads to greater action details when people are telling the truth vs. lying, but this is not the case when a Model Sketch sample is absent. But this seems a little misleading, in that there actually is a non-significant (p = .049) difference between truth-tellers and liars in the same direction in the Model Sketch-absent condition. (The authors use a conservative alpha level of .007, rather than .05, due to the multiple analyses being conducted, and that certainly makes sense.) But is the difference between the Model Sketch-present and Model Sketch-absent statistically significant? If so, that could be a little more clear. (Simply by eyeballing the data in Table 5, the pattern for the Model Sketch-present and Model Sketch-absent conditions look remarkably similar.)
The authors mention that their power analysis indicated that a sample of 162 participants should be adequate (bottom of p. 3). However, this is based on an assumption of a “medium to large effect size.” I’m not sure that the hypothesized effects in this study are medium to large sized; some effects of using a Model Sketch may be small, but nevertheless are important. Furthermore, the 2 x 3 between-participants design means that there were only 24 to 30 participants per condition (p. 4). I believe it is worth mentioning in the Discussion that a larger sample may provide more clarity in future studies.
Another key question for future research is the effect of Model Sketches on veracity judgments by multiple amateur judges. In other words, would judges’ normally very low deception detection rates across most study paradigms (usually around 54%) be improved when viewing senders who sketched their account of events while viewing a Model Sketch? This is worth mentioning briefly in the Discussion section as a direction for future research.
I have just a couple other minor issues to mention:
- Hypothesis 2 stated that participant in the Model Sketch-present condition would report the most PLATO details. It should be noted that the instructions for the Model Sketch-present condition actually include the instruction to “try to include as many details as possible in your sketch and description.” Was this sentence also included in the Model Sketch-absent condition? If not, couldn’t this have been an alternative explanation for why Hypothesis 2 would have been supported? (Since Hypothesis 2 received only partial support, I have listed this as a minor issue.)
- At the very end of p. 6, the authors could provide a little more detail regarding the methodology used to code open responses for participants’ strategies to appear convincing (particularly since inter-rater agreement was a little lower for this variable than for the variables mentioned in the preceding paragraphs).
- It would be helpful to briefly remind the reader which hypothesis is being tested by which analysis throughout the Results section – particularly on p. 8.
- I think a word may be missing from the sentence at the very top of p. 13 (line 517).
In sum, this manuscript adds substantively to the literature on investigative interview techniques for distinguishing between truthful and deceptive communications. Although not all hypotheses were supported, the pre-registered design increases confidence in the findings; Model Sketches are potentially a useful tool for deception detection, and this study helps elucidate when such sketches may be most useful. The authors should clarify which of the analyses were part of the pre-registered hypotheses and which analyses were exploratory. Also, the literature review would benefit from briefly discussing other researchers’ findings on the differences between truthful and deceptive communications.
Reviewer 2 Report
It was with great interest that I read the manuscript entitled: “The Model Sketch for Enhancing Lie Detection and Eliciting Information” (Manuscript ID brainsci-1884466). In this study, the authors investigated the effects of introducing a Model Sketch (an example of a detailed sketch) in investigative interviews on memory recall. For this purpose, 163 participants were asked to report a true or a false trip in two different phases of an interview. In phase one, a subgroup of participants performed a free recall, another subgroup was invited to sketch and narrate with Model Sketch, and a third subgroup did the same but without Model Sketch. In phase two, all participants made a free recall with no Model Sketch. The authors found evidence in favor of Model Sketch as a promising technique to aid information recall, particularly location-related information, and to help distinguish between truthful and deceitful accounts.
The study is pre-registered, which is a big plus, the methodology appears very sound, and it is great fitting to the Special Issue "Cognitive Approaches to Deception Research". Importantly, this one of the few available studies testing Model Sketch as an interview technique, thus being an important contribution to this field of research with important practical implications. Despite these valuable contributions, I leave here some comments, suggestions, and questions:
Introduction (starting page 2)
The introduction presents relevant literature regarding the main interview techniques that are the focus of the study: Sketch, Model Statement and, especially, Model Sketch. It encompasses all the information necessary to understand very well the study proposal. Also, there is a clear statement of the goals and hypotheses. I only thought of one suggestion:
- Considering this is a neuroscience journal and that the Special Issue Information states: “Both theoretical and empirical papers are welcome, focusing on observing behaviour, analysing speech content, measuring physiological responses and measuring brain activity.”, I was expecting as a reader some allusion to possible neural mechanisms underlying the memory benefits of the interview techniques under study (perhaps the studies discussing the neural correlates of drawing could be a nice parallel) or even some differences identified between lie and truth tellers.
Materials and Methods (starting page 3)
- Participants and design (page 3) - Considering that there was an ethnicity analysis to confirm that this variable did not affect the results, was something similar done for age (given the wide range)? This is also an important sociodemographic variable to consider as episodic memory is significantly modulated by age (e.g., Davison et al., 2006 - https://doi.org/10.3758/CABN.6.4.306; Spencer & Raz, 1995 - https://doi.org/10.1037/0882-7974.10.4.527). And in the case of sex (given there is evidence showing an advantage for females in episodic memory tasks that involve verbal abilities; Asperholm et al., 2019 - https://doi.org/10.1037/bul0000197)?
- Relatedly, while reading the eligibility criteria, it was not clear for me if there was some type of screening of health-related factors that might influence memory performance (e.g., level of fatigue; sleep problems and disorders; presence of cardiac/respiratory/hepatic medical conditions; history of developmental disorders; history of neurological/psychiatric disorders).
- Could the authors provide more information about the process of assigning the participants to the experimental conditions? How was the randomization process (e.g., was the randomization stratified according to some sociodemographic factor)?
Results (starting page 7)
- Post-Interview Questionnaire (page 7) – Considering that the dependent variables were measured on a 7-point scale (except truthfulness) and tested one at a time (from my understanding), did the authors contemplate the possibility of using non-parametric tests (e.g., Kruskal–Wallis H tes) given the ordinal nature of the items? This same question extends to the analyses conducted with the subset of participants that evaluated the usefulness of sketching.
Discussion (starting page 11)
The discussion raises important points and it brings together the key findings of the study in a brief and consistent manner. I leave here two small comments for the authors’ appreciation:
- For the same reasons aforementioned, the inclusion of some neuroscientific evidence that might supplement the findings would be nice.
- The authors could elaborate further on the limitations and strenghts of the study.
Minor comments:
- The text is very clear and easy to follow.
- Page 9 – There is an unnecessary space between “(…) truth tellers” and “provided more action details (…)”. In Table 3, there is an additional line in the bottom border of the text “Phase 1”. In Table 4, the text “Free call” is not aligned with the other subheadings.
- Regarding the citations and reference list, there are just a few minimal inconsistencies in the “doi” reporting. For example, in the reference (8), the doi link has an additional “dx”. Apparently, the “doi” is missing for reference (10) and reference (47).
Thank the authors for their contribution and best wishes for their work.
Sincerely,
